# CL-Informer: Long time series prediction model based on continuous wavelet transform

**Baijin Liu** [1]*, **Zimei Li**[2], **Zhanlin Li**[1], **Cheng Chen**[1]

**1** Jilin Institute of Chemical Technology, Longtan, Jilin, Jilin, China, **2** Changchun Institute of Technology, Chaoyang, Changchun, Jilin, China

* 1692797200@qq.com

**Data Availability Statement:** All experimental data files are available from figshare at https://doi.org/10.6084/m9.figshare.25498354.v1.

**Funding:** The author(s) received no specific funding for this work.

## Abstract

Time series, a type of data that measures how things change over time, remains challenging to predict. In order to improve the accuracy of time series prediction, a deep learning model CL-Informer is proposed. In the Informer model, an embedding layer based on continuous wavelet transform is added so that the model can capture the characteristics of multi-scale data, and the LSTM layer is used to capture the data dependency further and process the redundant information in continuous wavelet transform. To demonstrate the reliability of the proposed CL-Informer model, it is compared with mainstream forecasting models such as Informer, Informer+, and Reformer on five datasets. Experimental results demonstrate that the CL-Informer model achieves an average reduction of 30.64% in MSE across various univariate prediction horizons and a reduction of 10.70% in MSE across different multivariate prediction horizons, thereby improving the accuracy of Informer in long sequence prediction and enhancing the model's precision.

## Introduction

Early research on the TSF problem was primarily based on classical mathematical models and algorithms rooted in statistical principles and assumptions, such as auto-regressive (AR) [1], moving average (MA) [2], autoregressive moving average (ARIMA) [3], seasonal autoregressive moving average (SARIMA) models [4], among others. These models assume stationarity of data and capture autocorrelation, moving averages, and seasonality by establishing lag values and residuals for time series data. However, traditional statistical methods are typically built upon linear model assumptions. In real-world time series data, non-linear trends, seasonality, and other intricate patterns may not be effectively captured by conventional statistical approaches.

In order to enhance the accuracy of time series predictions, machine learning-based methods have been extensively employed. Recurrent neural network (RNN) is a suitable neural network for processing sequences [5]; However, it encounters issues such as gradient vanishing, gradient exploding, and limited parallelism [6]. Long Short-Term Memory Network (LSTM) partially addresses the problems of gradient vanishing and exploding in RNN by

**Competing interests:** The authors have declared that no competing interests exist.

incorporating gate mechanisms and cell states, enabling RNN to capture certain levels of long-term dependencies within the processed sequence. Nevertheless, LSTM still confronts challenges in capturing and reducing long-term dependencies when dealing with extensive periods or complex dependency relationships within the sequence. The literature introduces a time series modeling method based on Convolutional Neural Networks (CNN) [7], known as TCN. While TCN captures local patterns and dependencies through convolution operations on time series data, its model structure may not adequately capture long-term dependencies since TCN's convolution operations primarily focus on local neighborhoods [8]. Neither the RNN-based nor TCN-based model explicitly models distant temporal dependencies or facilitates efficient information exchange between them.

The Transformer model (Vaswani et al., 2017) is a neural network architecture based on self-attention mechanism [9], initially designed for natural language processing tasks like machine translation. It eliminates the sequential processing constraint and enables parallelization of sequence data processing. While self-attention has shown significant effectiveness in capturing dependencies among each element, the computational complexity grows quadratically when dealing with sequences. Various self-attention mechanisms have been proposed to address this issue in recent years. LogSparse Transformer (Li et al., 2019) introduced LogSparse self-attention, which breaks the memory bottleneck by selecting elements at exponentially growing intervals [10]. Performer (2020) is a Transformer-based acceleration model that utilizes a low-rank attention mechanism and random feature mapping to reduce computation and storage complexity [11]. Nyströmformer explores linear attention in a dual softmax form, which reduces the computational complexity of the self-attention mechanism by employing low-rank matrix approximation [12], thus improving the scalability and efficiency of the model to some extent. However, the Nyström method approximates large kernel matrices, which may increase computational complexity when dealing with large datasets or long sequences. Informer (Zhou et al., 2021) introduces sparse self-attention to replace the conventional self-attention [13], achieving time complexity of O(Llog L) and memory usage of O (Llog L) in dependency alignment. Although Informer demonstrates outstanding performance in capturing long-range dependencies with its self-attention mechanism, using distillation methods for model performance improvement and compression can lead to the loss and weakening of long-term dependencies in temporal sequences. Informer is also a Transformer-based neural network, so its ability to capture local dependencies is relatively weaker.

We propose CL-Informer, which is an improved hybrid model based on Informer. In order to improve Informer's ability to capture local and long dependencies, we have improved the model in two aspects, adding an embedding layer of continuous wavelet transform to the encoding part of the input sequence, adding two layers of LSTM after the self-attention module and deleting the 'distillation' method [14]. While distillation operations improve the model's training efficiency and generalization ability, they may affect the LSTM's capture of long-term dependencies in time series, thus reducing the model's predictive performance. By improving local dependence and long dependence, the model can comprehensively capture the correlation of different scales and ranges in time series and improve the data's modeling ability and prediction accuracy. This will make the model more adaptable to various complex time series data, leading to better performance and more accurate results in practical applications.

The main contributions of this paper are as follows:

1. We first design an embedded layer CWT based on continuous wavelet transform, through which the model can better learn and process data of different scales and improve model prediction accuracy.

2. We add the LSTM layer after the sparse self-focusing blocks of the coding layer and decoder to further capture the long and local dependencies of the model. The long-term forecasting model CL-Informer based on CWT and LSTM is established.

3. Many experiments show that our CL-Informer model has better predictive performance in long-term series prediction.

## Proposed deep learning model

### Problem definition

Multivariate time series forecasting is a time series analysis method used to predict multiple future values or trends simultaneously. Typically, we have an input variable $X_t = \{x_{t_1}, \ldots, x_{t_0} | x_{t_i} \in R^{d_x}\}$, and our goal is to forecast the values $Y_t = \{y_{t_1}, \ldots, y_{t_0} | y_{t_i} \in R^{d_y}\}$ for multiple consecutive future time points based on past data. Here, $d_x$ and $d_y$ represent the dimensions of the input and output variables, respectively. The model for Long Short-Term Forecasting (LSTF) can be represented as Eq (1):

$$\hat{Y}_t = F(X_t, \Lambda) \tag{1}$$

The predicted time series denoted as $\hat{Y}_t$, is influenced by the hyperparameter $\Lambda$ in the prediction model.

### CL-Informer

The overall architecture of the CL-Informer model is similar to that of Informer, as shown in Fig 1. For long sequence time series forecasting tasks, we employ a two-stage process. The first stage is the Time Series Embedding Layer, and the second is the Encoder-Decoder. The Time Series Embedding Layer maps the original time series data into a low-dimensional continuous vector space. The encoder part of the model encodes the embedded representation of the time series data into a fixed-length vector, capturing the relationships between different time steps. Subsequently, the decoder part decodes the encoded vector into the predicted target sequence.

**Model input.** The way in which the original time series is inputted into CL-Informer differs from Informer. Informer places more emphasis on modeling time information. By introducing input embeddings and positional encoding, Informer encodes time series data to capture temporal and sequential information, allowing the model to perceive the order of time and local/global temporal dependencies within the sequence. In the Embedding layer of CL-Informer, we transform the time series data into time-frequency information using continuous wavelet transformation and encode it, enabling the model to capture more local and global dependencies.

As shown in Fig 2, a CWT coding layer is added to the time series embedding layer [15]. Continuous wavelet transform has the characteristics of multi-scale and time-frequency localization and has been widely used in signal processing and analysis. It can provide a more comprehensive characterization of time series, especially for time series analysis with non-stationarity, transient characteristics, or frequency changes. Specifically, the embedding layer based on time series includes position coding, time coding, input sequence value coding, and continuous wavelet transform coding. The calculation process can be written as Eq (2):

$$I = Position(X) + Time(T) + Value(X) + CWT(X) \tag{2}$$

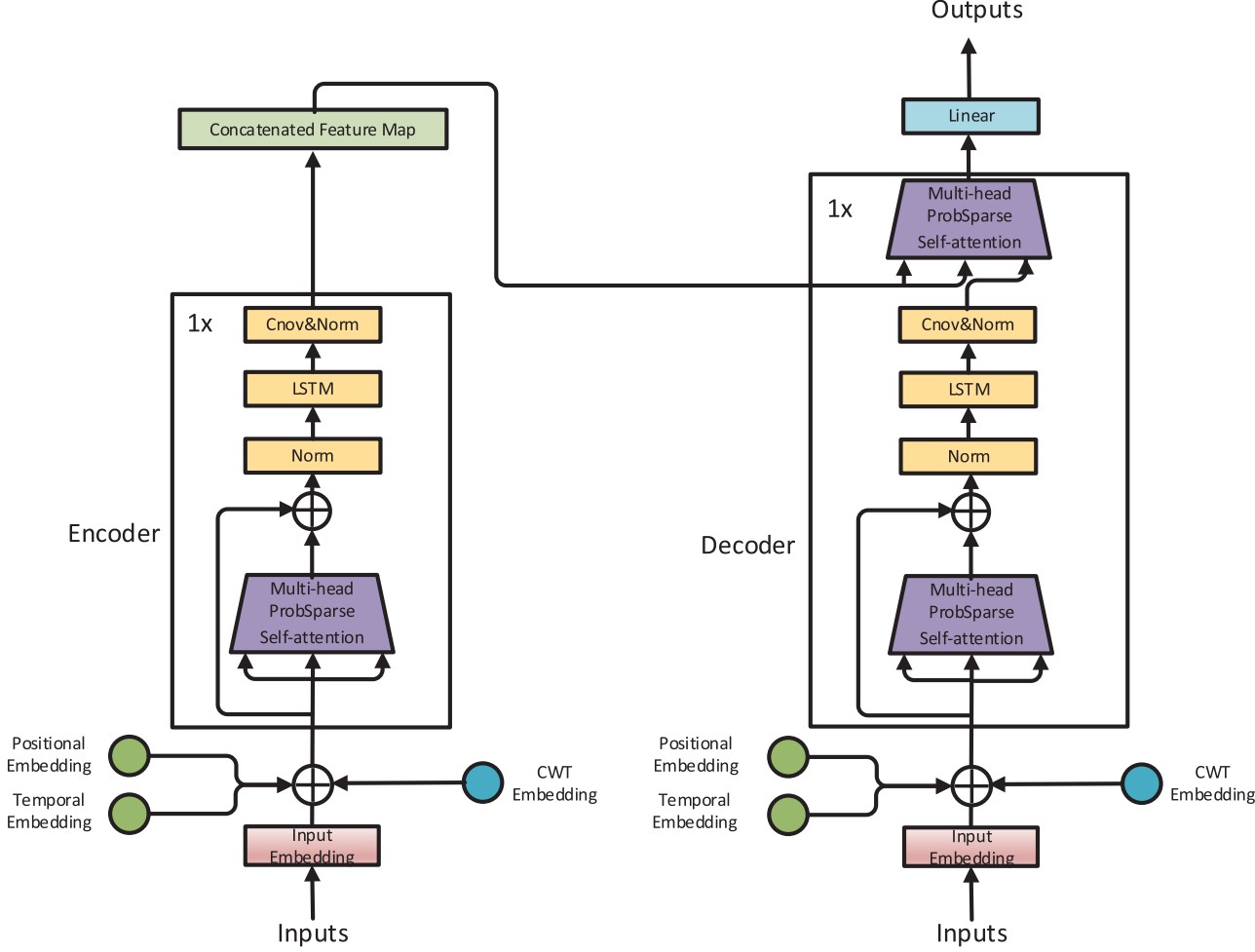

**Fig 1. Overall framework of CL-Informer.**

In the equation, $X$ represents the values of the input long time series, $T$ represents the time information generated by the time series, $Position(X)$ represents the positional encoding for the time series, $Time(T)$ represents the encoding for the time series values, $Value(X)$ represents the encoding for the input time series values, and $CWT(X)$ represents the encoding of the continuous wavelet transform for the time series.

The implementation process of the Continuous Wavelet Embedding layer is shown in Fig 3. In this process, $Cwt$ [16] represents the computation of the Continuous Wavelet Transform. In the Continuous Wavelet Transform, we use the 'Morlet' wavelet as the wavelet basis to compute the transformation of the input time series in both the time and frequency domains at multiple scales. The specific calculation can be derived from Eqs (3) and (4).

$$W_{a \times L_x} = Cwt(a, b) = \langle f, \Psi_{a,b} \rangle \frac{1}{\sqrt{a}} \int_{-\infty}^{+\infty} f(t) \Psi^* \left( \frac{t-b}{a} \right) \mathrm{d}t \tag{3}$$

$$\Psi(t) = e^{\frac{-t^2}{2}} e^{i\omega_0 t} \tag{4}$$

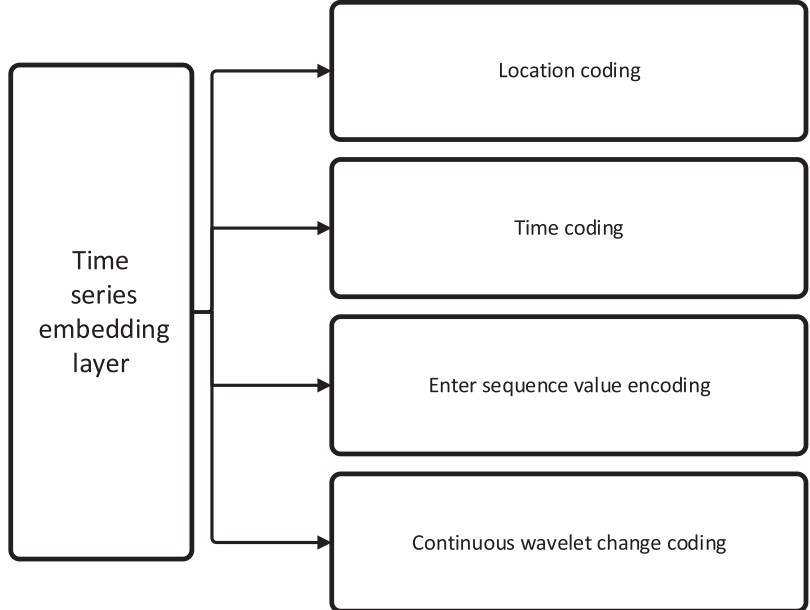

**Fig 2. The architecture of the time series embedding layer.**

In the given context, *a* represents the scale factor, *b* represents the translation factor, $f(t)$ denotes a function related to the time series, $\Psi(t)$ represents the expression of the 'Morlet' wavelet function, where $\omega_0$ signifies the central frequency. We compute the time series using formula *Cwt*, resulting in a coefficient matrix $W_{a \times L_x} \in R^{a*L_x}$ for the time series at multiple scales, with $L_x$ representing the length of the input sequence. $W_{a \times L_x} \in R^{a*L_x}$ is obtained through scale transformation analysis of the time-frequency characteristics of the time series at

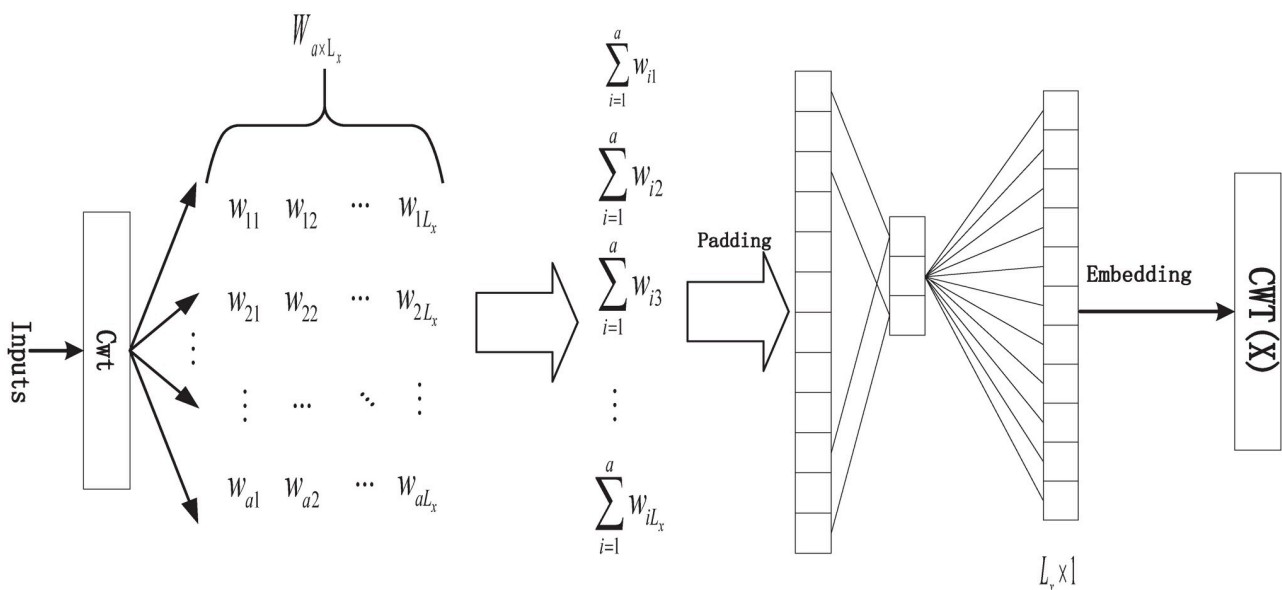

**Fig 3. CWT embedded layer architecture.**

different scales using the 'Morlet' wavelet as the central frequency. This matrix reflects the scale coefficients in the time-frequency domain, portraying the time series' variations across different spatial and frequency scales, thereby extracting both local structures and global features. Subsequently, we aggregate $W_{a \times L_x} \in R^{a*L_x}$ along the second dimension to merge the frequency domain features obtained at different scales within the same time domain, yielding $W_t$. Finally, the resulting multi-scale time-frequency domain features undergo encoding through a convolution block with a 3-sized kernel, as represented by Eqs (5) and (6), ultimately yielding the embedding layer $CWT(X)$ for the time series concerning continuous wavelets.

$$W_t = \left\{ \sum_{i=1}^{c} w_{i1}, \sum_{i=1}^{c} w_{i2}, \cdots, \sum_{i=1}^{c} w_{it} \right\} \tag{5}$$

$$CWT(X) = Embedding(Conv1d(Padding(W_t))) \tag{6}$$

However, by selecting the appropriate wavelet basis function, the continuous wavelet transform can capture the signal characteristics better and achieve accurate frequency analysis and time-frequency representation. In order to capture more time series features in the time-frequency domain, the transformer oil temperature dataset ETTh1 with a granularity of 1 hour and ETTm1 with a granularity of 15 minutes were selected for wavelet selection in this study. Both data sets were taken from the same transformer in a county in China and spanned two years.

This study selected three commonly used wavelet bases, Morlet, Gaus wavelet, and Marr wavelet, to verify their ability to capture temporal features in CL-Informer. For the Gaus wavelet, the eighth-level wavelet basis from the Gaussian wavelet series was chosen. To compare the feature capturing abilities of the three wavelet bases, the performance of MSE for single-element prediction was compared on the ETTh2 dataset with an input length of 168 and prediction lengths of {24, 48, 168, 336, 720}.

Fig 4 shows that compared with the Gaus8 wavelet base and Morlet wavelet base on the ETTh2 dataset, the Marr wavelet base is relatively less sensitive to the model. The influence of the Gaussian wavelet base and Morlet wavelet base on the mean square error of the model is similar. However, the support length of the Gaussian wavelet base is longer, and the computational complexity is higher. In order to reduce the computational complexity of the model, we choose different support lengths of Gaussian wavelet bases. The effects of MSE and MAE on the single element prediction performance were compared on the ETTh1 dataset with an input length of 168 and prediction length of 24 under different support lengths of Gaussian wavelet.

From Fig 5, it can be observed that as the support length of the Gaus wavelet increases, the prediction performance gradually improves. Among them, the Gaus6 wavelet basis exhibits the best prediction performance and reduces the wavelet transformation's complexity compared to the Gaus8 wavelet basis. Therefore, we will compare the Gaus6 wavelet basis with the Morlet wavelet basis in single-element and multi-element predictions on the ETTm1 dataset to select the wavelet basis with better prediction accuracy.

As can be seen from Fig 6, it can be observed that during training with an input length of 168 and prediction lengths of {24, 48, 96, 288, 672}, for the ETTm1 dataset with a granularity of 10 minutes, CL-Informer employed both Morlet and Gaus6 wavelets for continuous wavelet transform. In most cases, the Morlet wavelet transform model exhibited lower mean squared error (MSE) and mean absolute error (MAE) than the Gaus6 wavelet transform model. This indicates that the Morlet wavelet demonstrates better accuracy in single-element and multi-element predictions. Moreover, when conducting continuous transforms, the computational

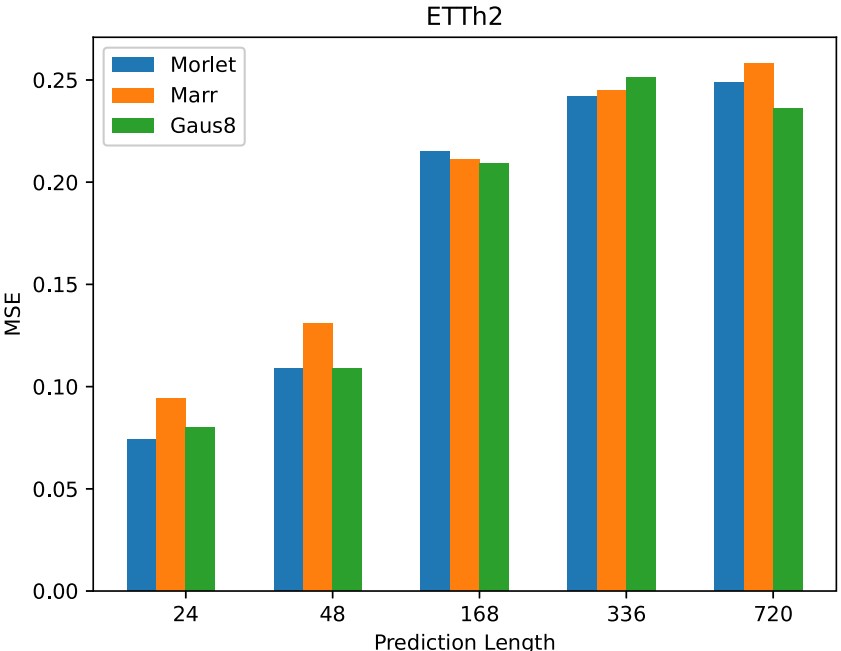

**Fig 4. The architecture of the time series embedding layer.**

complexity of the Morlet wavelet is lower than that of Gaus6. Therefore, this study selected the Morlet wavelet as the basis function for continuous transformation.

**Encoder-Decoder.** The main component of Informer relies on the self-attention mechanism, specifically the ProbSparse attention mechanism that reduces time complexity. ProbSparse attention creates a sparse matrix based on the KL divergence, selecting a few high-scoring dot

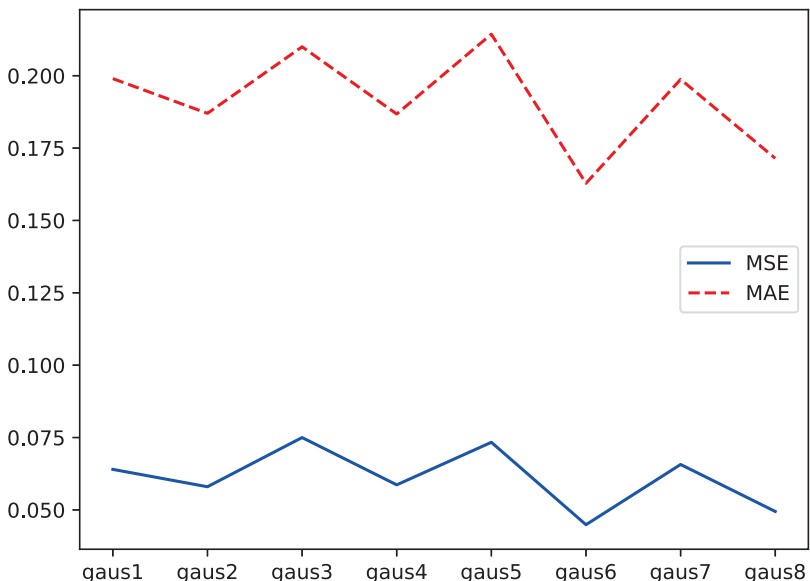

**Fig 5. Gaussian wavelet prediction of different support degree.**

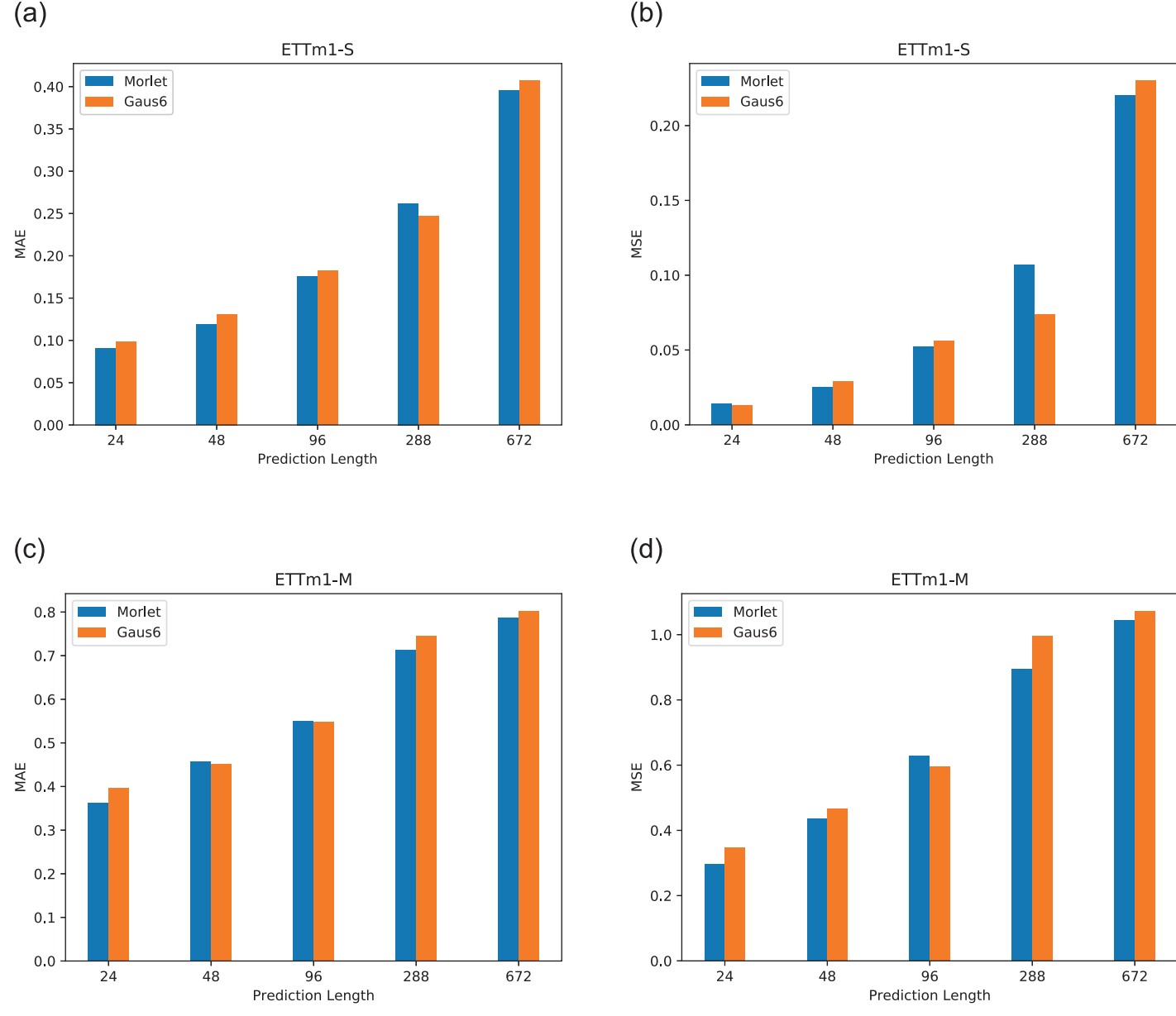

**Fig 6. Morlet and Gaus6 for single and multi-element prediction in ETTm1.**

products and taking the average of the other low-scoring dot products to reduce time complexity and memory usage. Compared to the regular Transformer model's self-attention mechanism, it achieves a complexity of O(LlogL). ProbSparse defines Q, K, and V as the query, key, and value vectors. However, it only includes the key-value pairs $\bar{Q} \in R^{L_Q \times d}$ under the sparse query Top-u, where the size of you is determined by a sampling parameter, and $d$ represents the corresponding dimensionality. The definition of ProbSparse is shown in Eq (7).

$$A(\bar{Q}, K, V) = Softmax\left(\frac{\bar{Q}K^t}{\sqrt{d}}\right)V \tag{7}$$

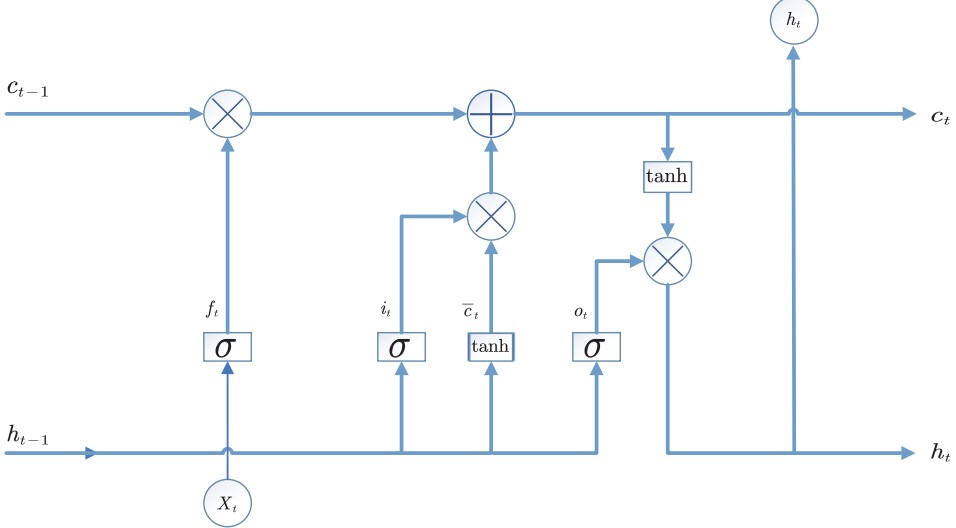

**Fig 7. LSTM memory unit.**

As shown in Fig 1, we also adopted the self-focusing mechanism of ProbSpare in the CL-Informer model, but we deleted the distillation module and added the LSTM layer. Because we add continuous wavelet transform to the model input to enhance the local and global dependence of the model input, it also brings a lot of noise and some redundant information to the model input. The LSTM layer can control the transfer of information through the gated unit, and the memory unit can effectively capture and store long-term memory and selectively transfer and forget information when needed.

As shown in Fig 7, LSTM [17] memory unit can selectively forget the information redundancy caused by continuous wavelet transform coding and further extract the long dependency of input time series so that the experimental model has better robustness. The definition of LSTM at time t is as follows:

$$f_t = \sigma(W_f[h_{t-1}; X_{it}] + b_f) \tag{8}$$

$$i_t = \sigma(W_i[h_{t-1}; X_{it}] + b_i) \tag{9}$$

$$o_t = \sigma(W_o[h_{t-1}; X_{it}] + b_o) \tag{10}$$

$$c_t = f_t \otimes c_{t-1} + i_t \otimes \tanh(W_c[h_{t-1}; X_{it}] + b_c) \tag{11}$$

$$h_t = o_t \otimes \tanh(c_t) \tag{12}$$

$$y_t = W_d h_t + b_d \tag{13}$$

In order to better learn the long-term dependency of LSTM, the output of the updated LSTM unit is represented as follows:

$$y_t = W_y y_t + W_{\bar{c}} c_t \tag{14}$$

Where $W_f$, $W_i$, $W_o$, $W_c$, $W_d$, $W_y$, $W_c$, $W\bar{c}$, $b_f$, $b_i$, $b_o$, $b_c$ and $b_d$ represent the learnable

parameters, $c_t$ and $h_t$ represent the storage cell state variable and hidden layer state variable, tanh represents the hyperbolic tangent activation function, and $\sigma$ represents the sigmoid activation function.

In the CL-Informer model, we use a 2-layer LSTM to capture further long-term dependencies in the time series, but to reduce computational complexity, we change the number of layers in the encoder to one, thereby reducing the spatial complexity of the calculations. Finally, the output $\hat{Y}$ is obtained through a fully connected layer for result prediction.

## Results

### Experiment

**Implementation step.** Using the ADAM optimizer with an initial learning rate of $1E - 4$, the model was trained for 10 epochs with a batch size of 32. The training process was terminated prematurely after 10 iterations. All experiments were repeated 10 times and conducted on an NVIDIA GTX4070 GPU with a RAM capacity of 12GB. CL-Informer differs from Informer and InformerStack in having only one encoder layer and an eight-head attention decoder layer. The prediction windows are consistent across all configurations based on the dataset's timestamps, while the input sequence length for the CL-Informer model encoder is set to 168, and the start mark (label length) for the decoder is fixed at 48. Two evaluation metrics, namely $MSE = \frac{1}{n}\sum_{i=1}^{n}(y - \hat{y})^2$ and $MAE = \frac{1}{n}\sum_{i=1}^{n}|y - \hat{y}|$, were employed for each prediction window, with a stride value of one applied to traverse through the entire dataset.

**Data.** To evaluate CL-Infomer, we conducted experiments using multiple datasets. In order to explore the long dependency enhancement effect of the model, we conducted experiments using five datasets mentioned in Zhou et al. (2021).

1. ETTh and ETTm are two different granularity datasets that contain the load characteris-tics of seven oil and power transformers, respectively, from July 2016 to July 2018. ETTh contains the two-hour dataset {$ETTh1$, $ETTh2$}, and ETTm contains the 15-minute dataset ETTm1.

2. The Wheather dataset contains local climate data for nearly 1,600 locations in the United States between 2010 and 2013. Data from each site was collected at an hourly rate for a total of four years. The data set contains the target value "wet bulb number" and 11 climate characteristics.

3. The ECL dataset collected hourly electricity consumption from 321 customers from 2012 to 2014. We split the data set according to Zhou et al(2021). For the ETT data set, the train-ing/validation/test set contains data for 12/4/4 months. For the Wheather data set, the train-ing/cycle/test was 28/10/10 months and the ECL was split according to the train-ing/cycle/test was 15/3/4 months.

**Datum line.** In time series prediction, we chose six models to compare with CL-Informer, including two RNN-based models, LSTMa (Bahdanau et al., 2015) [18] and LSTnet (Lai et al., 2018) [19]. DeepAR [20], a neural network based on autoregressive loops (Flunkert et al., 2017), and transformer-based Reformer (Kitaev, Kaiser, and 2019) [21], LogSparses(Li et al., 2019), Informer (Zhou et al., 2021). In order to explore the improvement of the accuracy of CL-Informer's prediction of long-time series better, we added a normalized self-attention variant (Informer+) to the experiment for comparison.

## Results and analysis

As shown in Fig 8, with an input length of 168, as the prediction lengths extend to {24, 48, 96, 288, 672}, the CL-Informer model exhibits lower mean squared error (MSE) and mean absolute error (MAE) values compared to the Informer model in both single-element and multi-element predictions on the ETTm1 dataset. Therefore, we have enhanced the model's prediction accuracy by introducing measures such as the CWT embedding layer, eliminating distillation methods, and incorporating LSTM.

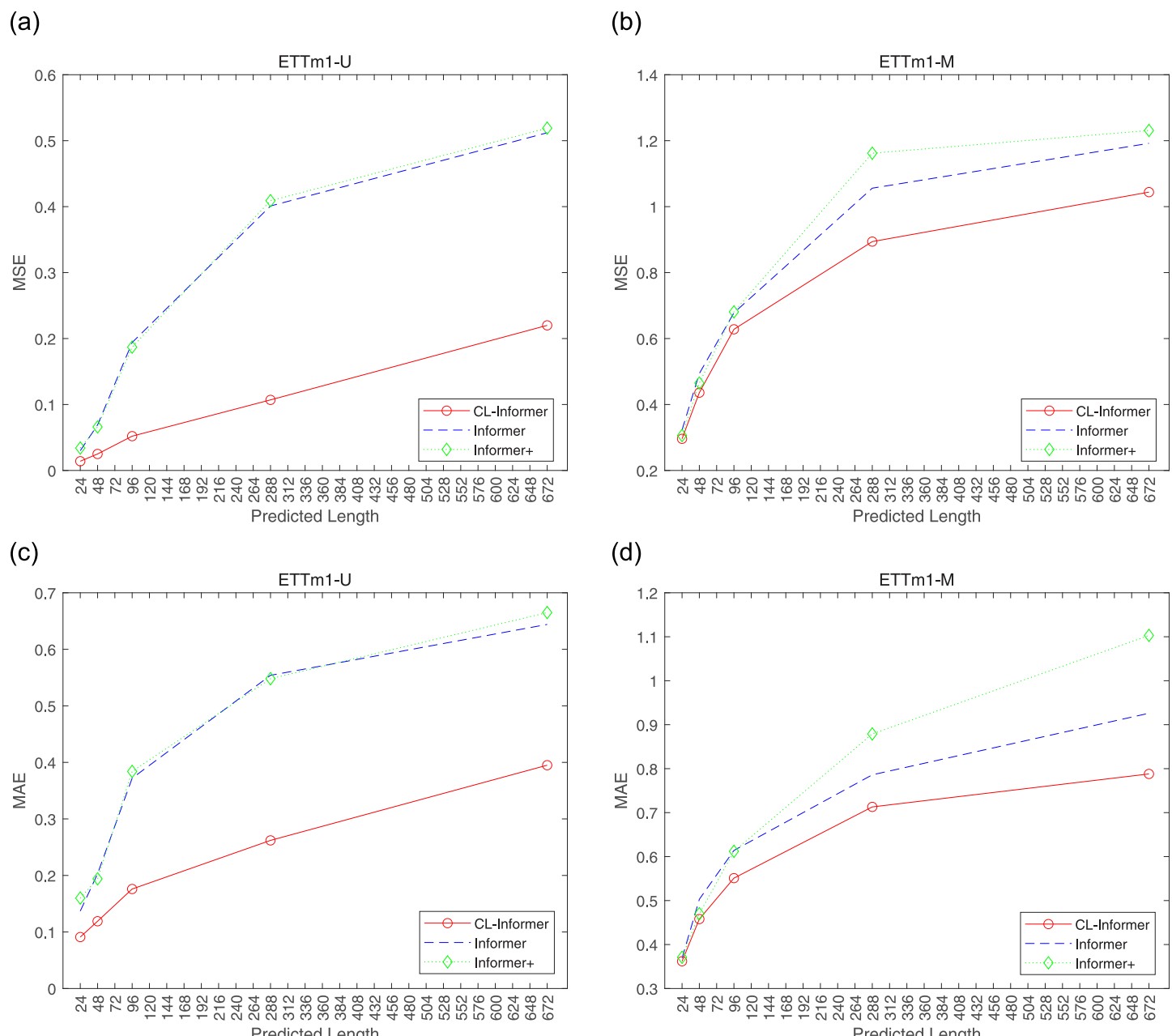

**Fig 8. MSE and MAE graphs of different prediction lengths of CL-Informer and Informer under ETTm1.**

To thoroughly compare CL-Informer and Informer and other time series models, we conducted comprehensive multi-element and single-element prediction experiments on different data sets and in different Settings. For all data sets, the input length is 168, where ETTh1, ETTh2, Wheather prediction length includes {24, 48, 168, 336, 720}, ETTm1 prediction length includes {24, 48, 96, 288, 672}, The ECL prediction length is selected from {48, 168, 336, 720, 960}. They are highlighted in bold to show the best results for different forecast periods.

**Cell time prediction.** Table 1 shows that CL-Informer outperforms all models except for DeepAR in all cases and only slightly performs worse on a few datasets. For example, in the input-168-predict-720 setting, compared to the state-of-the-art results, CL-Informer achieves a 37.9% reduction in MSE for ETTh1 (0.269→0.160), a 10.2% reduction for ETTh2 (0.277→0.249), a 34.8% reduction for Weather (0.359→0.234), and a 37.1% reduction for ECL (0.540→0.385). Additionally, in the input-168-predict-672 setting, CL-Informer achieves a 57.0% reduction in MSE for ETTm1 compared to the state-of-the-art results (0.512→0.220). CL-Informer exhibits an overall MSE reduction of 35.4% in the mentioned settings.

Furthermore, we can observe that CL-Informer demonstrates good stability as the prediction length increases, particularly in longer prediction horizons. This indicates that CL-Informer is more suitable for long-term time series forecasting, which is significant for real-world applications like long-term energy consumption planning and weather forecasting.

**Table 1. Data set (5 cases) single element long series time prediction results.**

| Methods | | CL-Informer | | Informer | | Informer+ | | LogTrans | | Reformer | | LSTMa | | DeepAR | |
|---|---|---|---|---|---|---|---|---|---|---|---|---|---|---|---|
| Metric | | MSE | MAE | MSE | MAE | MSE | MAE | MSE | MAE | MSE | MAE | MSE | MAE | MSE | MAE |
| ETTh1 | 24 | **0.069** | **0.206** | 0.098 | 0.247 | 0.092 | 0.246 | 0.103 | 0.259 | 0.222 | 0.389 | 0.114 | 0.272 | 0.107 | 0.280 |
| | 48 | **0.090** | **0.242** | 0.158 | 0.319 | 0.161 | 0.322 | 0.167 | 0.328 | 0.284 | 0.445 | 0.193 | 0.358 | 0.162 | 0.327 |
| | 168 | **0.096** | **0.241** | 0.183 | 0.346 | 0.187 | 0.355 | 0.207 | 0.375 | 1.522 | 1.191 | 0.236 | 0.392 | 0.239 | 0.422 |
| | 336 | **0.083** | **0.221** | 0.222 | 0.387 | 0.215 | 0.369 | 0.230 | 0.398 | 1.860 | 1.124 | 0.590 | 0.698 | 0.445 | 0.552 |
| | 720 | **0.160** | **0.325** | 0.269 | 0.435 | 0.257 | 0.421 | 0.273 | 0.463 | 2.112 | 1.436 | 0.683 | 0.768 | 0.658 | 0.707 |
| ETTh2 | 24 | **0.074** | **0.209** | 0.093 | 0.240 | 0.099 | 0.241 | 0.102 | 0.255 | 0.263 | 0.437 | 0.155 | 0.307 | 0.098 | 0.263 |
| | 48 | **0.109** | **0.263** | 0.155 | 0.314 | 0.159 | 0.317 | 0.169 | 0.348 | 0.458 | 0.545 | 0.190 | 0.348 | 0.163 | 0.341 |
| | 168 | **0.215** | **0.371** | 0.232 | 0.389 | 0.235 | 0.390 | 0.246 | 0.422 | 1.029 | 0.879 | 0.385 | 0.514 | 0.255 | 0.414 |
| | 336 | **0.242** | **0.401** | 0.263 | 0.417 | 0.258 | 0.423 | 0.267 | 0.437 | 1.668 | 1.228 | 0.558 | 0.606 | 0.604 | 0.607 |
| | 720 | **0.249** | **0.404** | 0.277 | 0.431 | 0.285 | 0.442 | 0.303 | 0.493 | 2.030 | 1.721 | 0.640 | 0.681 | 0.429 | 0.580 |
| ETTm1 | 24 | **0.014** | **0.091** | 0.030 | 0.137 | 0.034 | 0.160 | 0.065 | 0.202 | 0.095 | 0.228 | 0.121 | 0.233 | 0.091 | 0.243 |
| | 48 | **0.025** | **0.119** | 0.069 | 0.203 | 0.066 | 0.194 | 0.078 | 0.220 | 0.249 | 0.390 | 0.305 | 0.411 | 0.219 | 0.362 |
| | 96 | **0.052** | **0.176** | 0.194 | 0.372 | 0.187 | 0.384 | 0.199 | 0.386 | 0.920 | 0.767 | 0.287 | 0.420 | 0.364 | 0.496 |
| | 288 | **0.107** | **0.262** | 0.401 | 0.554 | 0.409 | 0.548 | 0.411 | 0.572 | 1.108 | 1.245 | 0.524 | 0.584 | 0.948 | 0.795 |
| | 672 | **0.220** | **0.395** | 0.512 | 0.644 | 0.519 | 0.665 | 0.598 | 0.702 | 1.793 | 1.528 | 1.064 | 0.873 | 2.437 | 1.352 |
| Weather | 24 | **0.091** | **0.208** | 0.117 | 0.251 | 0.119 | 0.256 | 0.136 | 0.279 | 0.231 | 0.401 | 0.131 | 0.254 | 0.128 | 0.274 |
| | 48 | **0.125** | **0.251** | 0.178 | 0.318 | 0.185 | 0.316 | 0.206 | 0.356 | 0.328 | 0.423 | 0.190 | 0.334 | 0.203 | 0.353 |
| | 168 | **0.214** | **0.339** | 0.266 | 0.398 | 0.269 | 0.404 | 0.309 | 0.439 | 0.654 | 0.634 | 0.341 | 0.448 | 0.293 | 0.451 |
| | 336 | **0.244** | **0.372** | 0.297 | 0.416 | 0.310 | 0.422 | 0.359 | 0.484 | 1.792 | 1.093 | 0.456 | 0.554 | 0.585 | 0.644 |
| | 720 | **0.234** | **0.375** | 0.359 | 0.466 | 0.361 | 0.471 | 0.388 | 0.499 | 2.087 | 1.534 | 0.866 | 0.809 | 0.499 | 0.596 |
| ECL | 48 | 0.235 | 0.359 | 0.239 | 0.359 | 0.238 | 0.368 | 0.280 | 0.429 | 0.971 | 0.884 | 0.493 | 0.539 | **0.204** | **0.357** |
| | 168 | 0.340 | **0.424** | 0.447 | 0.503 | 0.442 | 0.514 | 0.454 | 0.529 | 1.671 | 1.587 | 0.723 | 0.655 | **0.315** | 0.436 |
| | 336 | **0.344** | **0.430** | 0.489 | 0.528 | 0.501 | 0.552 | 0.514 | 0.563 | 3.528 | 2.196 | 1.212 | 0.898 | 0.414 | 0.519 |
| | 720 | **0.385** | **0.466** | 0.540 | 0.571 | 0.543 | 0.578 | 0.558 | 0.609 | 4.891 | 4.047 | 1.511 | 0.966 | 0.563 | 0.595 |
| | 960 | **0.367** | **0.457** | 0.582 | 0.608 | 0.594 | 0.638 | 0.624 | 0.645 | 7.019 | 5.105 | 1.545 | 1.006 | 0.657 | 0.683 |
| count | | 47 | | 0 | | 0 | | 0 | | 0 | | 0 | | 3 | |

**Table 2. Data set (4 cases) multi-element long series time prediction results.**

| Methods | | CL-Iformer | | Informer | | Informer+ | | LogTrans | | Reformer | | LSTMa | | LSTnet | |
|---|---|---|---|---|---|---|---|---|---|---|---|---|---|---|---|
| Metric | | MSE | MAE | MSE | MAE | MSE | MAE | MSE | MAE | MSE | MAE | MSE | MAE | MSE | MAE |
| ETTh1 | 24 | **0.546** | **0.533** | 0.577 | 0.549 | 0.620 | 0.577 | 0.686 | 0.604 | 0.991 | 0.754 | 0.650 | 0.624 | 1.293 | 0.901 |
| | 48 | **0.657** | **0.608** | 0.685 | 0.625 | 0.692 | 0.671 | 0.766 | 0.757 | 1.313 | 0.906 | 0.702 | 0.675 | 1.456 | 0.960 |
| | 168 | **0.881** | **0.747** | 0.931 | 0.752 | 0.947 | 0.797 | 1.002 | 0.846 | 1.824 | 1.138 | 1.212 | 0.867 | 1.997 | 1.214 |
| | 336 | **1.068** | **0.806** | 1.128 | 0.873 | 1.094 | 0.813 | 1.362 | 0.952 | 2.117 | 1.280 | 1.424 | 0.994 | 2.655 | 1.369 |
| | 720 | **1.209** | 0.901 | 1.215 | **0.896** | 1.241 | 0.917 | 1.397 | 1.291 | 2.415 | 1.520 | 1.960 | 1.322 | 2.143 | 1.380 |
| ETTm1 | 24 | **0.296** | **0.362** | 0.323 | 0.369 | 0.306 | 0.371 | 0.419 | 0.412 | 0.724 | 0.607 | 0.621 | 0.629 | 1.968 | 1.170 |
| | 48 | **0.436** | **0.458** | 0.494 | 0.503 | 0.465 | 0.470 | 0.507 | 0.583 | 1.098 | 0.777 | 1.392 | 0.939 | 1.999 | 1.215 |
| | 96 | **0.628** | **0.551** | 0.678 | 0.614 | 0.681 | 0.612 | 0.768 | 0.792 | 1.433 | 0.945 | 1.339 | 0.913 | 2.762 | 1.542 |
| | 288 | **0.894** | **0.713** | 1.056 | 0.786 | 1.162 | 0.879 | 1.462 | 1.320 | 1.820 | 1.094 | 1.740 | 1.124 | 1.257 | 2.076 |
| | 672 | **1.044** | **0.788** | 1.192 | 0.926 | 1.231 | 1.103 | 1.669 | 1.461 | 2.187 | 1.232 | 2.736 | 1.555 | 1.917 | 2.941 |
| Weather | 24 | **0.329** | **0.377** | 0.335 | 0.381 | 0.349 | 0.397 | 0.435 | 0.477 | 0.655 | 0.583 | 0.546 | 0.570 | 0.615 | 0.545 |
| | 48 | 0.410 | 0.437 | 0.395 | 0.459 | **0.386** | **0.433** | 0.426 | 0.495 | 0.729 | 0.666 | 0.829 | 0.677 | 0.660 | 0.589 |
| | 168 | **0.585** | **0.558** | 0.608 | 0.567 | 0.613 | 0.582 | 0.727 | 0.671 | 1.318 | 0.855 | 1.038 | 0.835 | 0.748 | 0.647 |
| | 336 | **0.639** | **0.589** | 0.702 | 0.620 | 0.707 | 0.634 | 0.754 | 0.670 | 1.930 | 1.167 | 1.657 | 1.059 | 0.782 | 0.683 |
| | 720 | **0.634** | **0.593** | 0.831 | 0.731 | 0.834 | 0.741 | 0.885 | 0.773 | 2.726 | 1.575 | 1.536 | 1.109 | 0.851 | 0.757 |
| ECL | 48 | **0.248** | **0.349** | 0.344 | 0.393 | 0.334 | 0.399 | 0.355 | 0.418 | 1.404 | 0.999 | 0.486 | 0.572 | 0.369 | 0.445 |
| | 168 | **0.273** | **0.364** | 0.368 | 0.424 | 0.353 | 0.420 | 0.368 | 0.432 | 1.515 | 1.069 | 0.574 | 0.602 | 0.394 | 0.476 |
| | 336 | **0.286** | **0.380** | 0.381 | 0.431 | 0.381 | 0.439 | 0.373 | 0.439 | 1.601 | 1.104 | 0.886 | 0.795 | 0.419 | 0.477 |
| | 720 | **0.312** | **0.399** | 0.406 | 0.443 | 0.391 | 0.438 | 0.409 | 0.454 | 2.009 | 1.170 | 1.676 | 1.095 | 0.556 | 0.565 |
| | 960 | **0.332** | **0.411** | 0.460 | 0.548 | 0.492 | 0.550 | 0.477 | 0.589 | 2.141 | 1.387 | 1.591 | 1.128 | 0.605 | 0.599 |
| count | | 37 | | 1 | | 2 | | 0 | | 0 | | 0 | | 0 | |

**Multi-element time prediction.** Table 2 shows the results of four typical datasets. Our model performs well in the long-term prediction task compared to other models. For example, in the input-168-predict-336 setting, MSE decreased by 5.6% (1.128→1.068) in ETTh1 and 8.9% (0.702→0.639) in Wheather compared with the previous most advanced result. The ECL was reduced by 24.9% (0.381→0.286), and the MSE of ETTm1 was reduced by 15.3% (1.056→0.894) in the input-168-predict-288 setting. Overall, the overall MSE of CL-Informer in the appellate setting decreased by 13.6%. We can find that the CL-Informer model's long time series prediction effect is better than that of a based time series model and some transformer-based time series models in multi-element time prediction.

**CL-Informer predicted effect.** The two pictures in Fig 9 respectively show the single-element prediction effect of CL-Informer under ETTm1. Since CL-Informer is a multi-step prediction model, we randomly select the predicted and actual values under a single step to compare. Fig 9(a) shows the actual and predicted values' curves in the ETTm1 data set under CL-Informer input-168-predict-288. Fig 9(b) shows the effect of the actual and predicted values in a single prediction window. From Fig 9, we can see that CL-Informer performs well predicting long series.

## Discussion

In order to verify the enhancement of the Informer model by CL-Informer, we also considered conducting additional ablation experiments on ETTh1 to compare the transformation of MSE and MAE of the C-Informer model after removing the LSTM coding layer. Tables 1 and 2 compare CL-Informer with Informer's model and the main-stream models LogTrans,

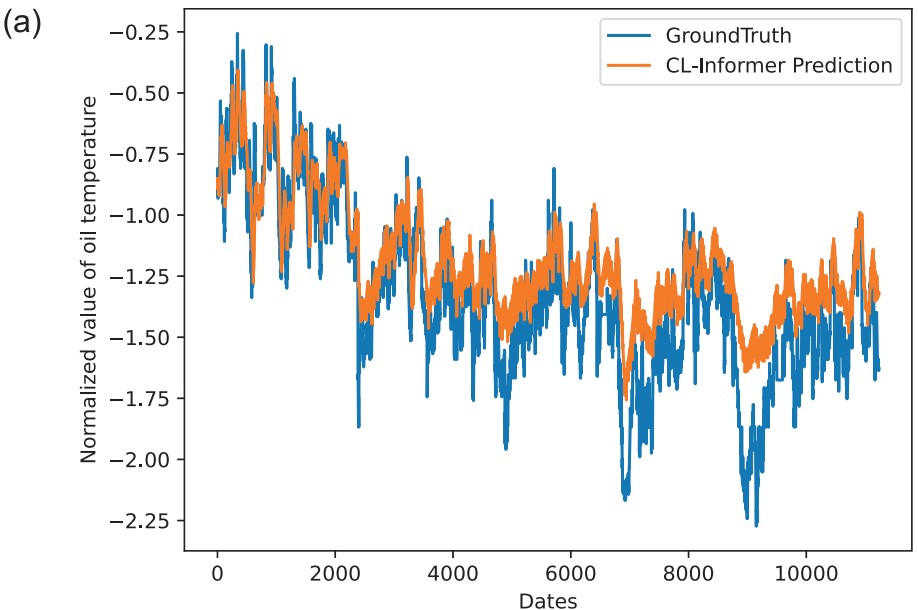

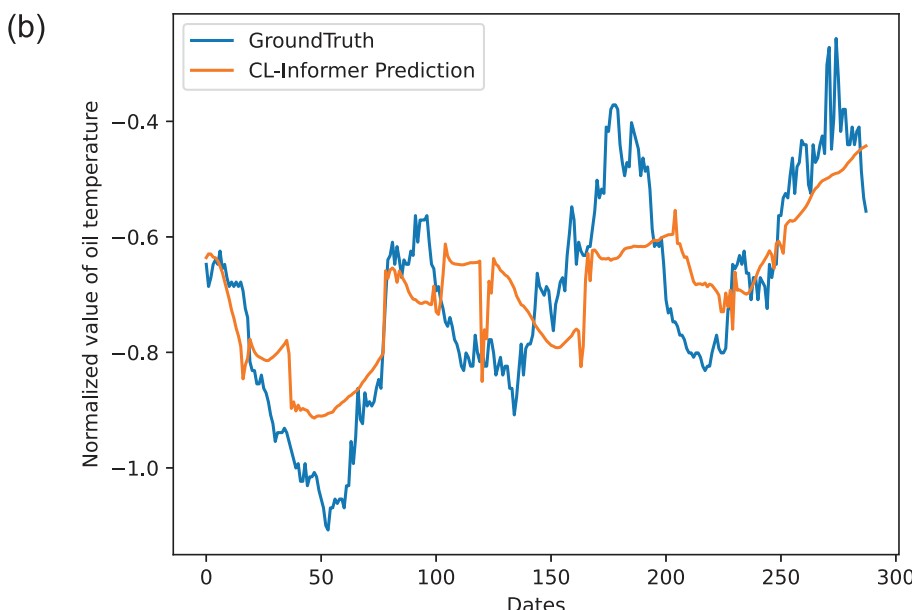

**Fig 9. Prediction effect slice of CL-Informer under ETTm1.**

Reformer, LSTMa and LSTnet. In this ablation experiment, we compare our approach to Informer's model. In Table 3, we compare the performance of CL-Informer, C-Informer, and Informer for MSE, MAE with input length {288, 576} and prediction length {720, 1440}. For a fair comparison, except for CL-Informer, C-Informer kept the encoder at layer 1, and the other hyperparameters were unchanged. In this experiment, data set ETTh1 was used for model ablation.

From Table 3, it can be observed that removing the LSTM layer resulted in inferior performance in terms of MSE and MAE compared to including the LSTM layer. The average MAE

**Table 3. Single element predictive ablation experiment of CL-Informer.**

| Prediction length | | 720 | | 1440 | |
|---|---|---|---|---|---|
| Encoder's input | | 288 | 576 | 288 | 576 |
| CL-Informer | MSE | 0.078 | 0.073 | 0.081 | 0.092 |
| | MAE | 0.225 | 0.216 | 0.228 | 0.24 |
| C-Informer | MSE | 0.096 | 0.08 | 0.098 | 0.098 |
| | MAE | 0.241 | 0.232 | 0.249 | 0.245 |
| Informer | MSE | 0.115 | 0.091 | 0.216 | 0.117 |
| | MAE | 0.269 | 0.235 | 0.395 | 0.272 |

increased by 6.4%, indicating poorer performance without the LSTM layer. Furthermore, the C-Informer, which only includes the CWT embedding layer, outperformed the Informer model in terms of MSE and MAE, with an average decrease in MAE of 20.6%. As the LSTM and CWT embedding layers are gradually removed, the model's performance continuously deteriorates. This indicates that by adding the CWT embedding layer along with the LSTM layer while removing distillation operations, the model's predictive accuracy is indeed improved. Fig 10 also demonstrates that as the CWT embedding layer and LSTM layer are added to the Informer model, the error consistently decreases, indicating that CL-Informer exhibits good predictive performance.

Although capturing more scale of time series features through continuous wavelet transform has enabled CL-Informer to outperform the Informer model in predictive accuracy across multiple datasets, further research is needed regarding selecting wavelet basis functions and extracting time-frequency domain features.

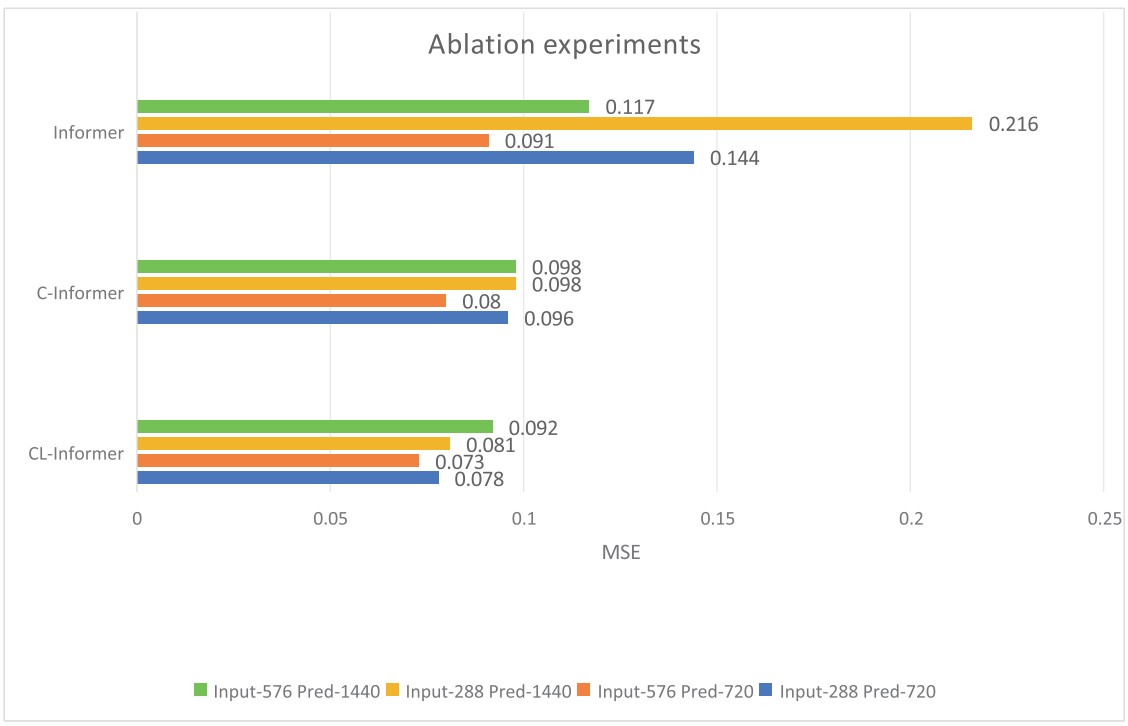

**Fig 10. Sensitivity of three wavelet bases to the model.**

## Conclusion

This paper studies the problem of long-series time series prediction, and an improved CL-Informer model for long-series prediction based on Informer is proposed. Specifically, to improve the model's prediction accuracy in long series, we embed a continuous wavelet transform layer (CWT) in the coding layer and extracted multi-scale features and dependencies from the time series through CWT. On this basis, we remove the "distillation" operation of the coding-decoder, add an extended short-term neural network (LSTM), change the number of encoder layers to 1 layer, and finally get the prediction model CL-Informer.

By comparing with other experimental models, it has been demonstrated that CL-Informer is more effective in discovering predictive dependencies. Additionally, under various experimental settings, CL-Informer consistently achieves better predictive performance. For single-variable prediction horizons, the CL-Informer model exhibited an average decrease of 30.64% in Mean Squared Error (MSE), while for multi-variable predictions, the reduction was 10.70%. This further validates the effectiveness and stability of the proposed model.

## Supporting information

**S1 Fig. CWT diagram of the embedding layer architecture.** Show the composition and execution flow of CWT embedding layer.
(VSDX)

**S2 Fig. Univariate linear comparison plot between MOrlet and Gaus6 wavelet basis in the ETTm1 dataset.** Discount plots of MSE versus MAE for both wavelet bases.
(VSDX)

**S3 Fig. Multivariate linear comparison plot between MOrlet and Gaus6 wavelet basis in the ETTm1 dataset.** Discount plots of MSE versus MAE for both wavelet bases.
(VSDX)

**S1 Table. CL-Informer compared to Informer in univariate accuracy optimization percentage.** Calculate the MSE and MAE improvement of CL-Informer compared with Informer in five data.
(XLSX)

**S2 Table. CL-Informer compared to Informer in multivariate accuracy optimization percentage.** Calculate the MSE and MAE improvement of CL-Informer compared with Informer in four data.
(XLSX)

**S1 File. The five kinds of data sets used in the experiments.**
(7Z)

## Acknowledgments

The author expresses gratitude to Zimei Li.

## Author Contributions

**Conceptualization:** Baijin Liu, Zimei Li.

**Data curation:** Baijin Liu, Zimei Li.

**Formal analysis:** Zimei Li.

**Methodology:** Baijin Liu, Zimei Li.

**Visualization:** Baijin Liu, Zhanlin Li, Cheng Chen.

**Writing – original draft:** Baijin Liu.

**Writing – review & editing:** Baijin Liu, Zimei Li.

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
