## [Decision Letter · Decision Letter 0]

26 Feb 2024

PONE-D-24-03744CL-Informer: Long Time Series Prediction Model Based on Continuous Wavelet TransformPLOS ONE

Dear Dr. Baijin Liu,

Thank you for submitting your manuscript to PLOS ONE. After careful consideration, we feel that it has merit but does not fully meet PLOS ONE’s publication criteria as it currently stands. Therefore, we invite you to submit a revised version of the manuscript that addresses the points raised during the review process.

We look forward to receiving your revised manuscript.

Kind regards,

Aamna Mohammed AlShehhi, PhD

Academic Editor

PLOS ONE

Journal Requirements:

"NO"

Reviewers' comments:

Reviewer's Responses to Questions

**Comments to the Author**

1. Is the manuscript technically sound, and do the data support the conclusions?

Reviewer #1: Yes

Reviewer #2: Partly

Reviewer #3: Partly

Reviewer #4: Yes

2. Has the statistical analysis been performed appropriately and rigorously? 

Reviewer #1: Yes

Reviewer #2: Yes

Reviewer #3: Yes

Reviewer #4: Yes

3. Have the authors made all data underlying the findings in their manuscript fully available?

Reviewer #1: Yes

Reviewer #2: Yes

Reviewer #3: Yes

Reviewer #4: Yes

4. Is the manuscript presented in an intelligible fashion and written in standard English?

Reviewer #1: Yes

Reviewer #2: Yes

Reviewer #3: Yes

Reviewer #4: No

5. Review Comments to the Author

Reviewer #1: The authors proposed CL-Informer, which is an improved hybrid model based on Informer. An embedded continuous wavelet transform layer in the encoder and decoder to improve the prediction accuracy of the model in long series. They added the LSTM layer after the sparse self-focusing blocks of the encoder and decoder to further capture the long and local dependencies of the model. Five experiments showed that the CL-Informer model has better predictive performance in long-term series prediction. This is a valuable job, but I still have some questions that need the authors to reply to.

1. Can the authors explain why they used those five time series datasets in this paper? CL-Informer can learn and process data of different scales, so, whether they can handle any datasets that are much larger than ETTh1, ETTh2, ETTm1, Weather, and ECL well?

2. Why in S2_Table includes ETTh2, but in S1_Table doesn’t?

3. In Figure 3, the x-axis is the prediction length in dataset ETTh2, which is {24, 48, 168, 336, 720}, but in the main manuscript content, it says the ETTh2 data set with input length of 168 and prediction length of {24, 48, 96, 168, 720}, which one is correct?

4. From Fig 4, the authors claimed as the support length of the Gaus wavelet increases, the prediction performance gradually improves. For example, MSE from 0.06 (gaus1) to 0.05 (gaus8) is only 0.01 improved, it seems like not a strongly improved.

5. Line 6 in paragraph one of the ‘Results and analysis’ part needs to be modified, that sentence is not finished.

6. Some figures can be merged as one figure, such as Figure 7 and Figure 8, the same datasets, but one is for MSE and the other is for MAE score, which can be put into one figure. The authors need to pay attention that in some figures they used MSE, but in others, they used MSE score.

7. It’s better to add some legends for the figures, like Figure 9, to make it more readable for the readers.

8. It’s better to let a native speaker go through this whole manuscript and modify the grammar issues.

Reviewer #2: The manuscript presents the CL-Informer model, an advancement over existing time series prediction models through the integration of a continuous wavelet transform embedding layer and an LSTM layer.

1. Quality of Fig 4,5,7, need improved

2. In the abstract and conclusion, it is necessary to introduce the superiority of CL-Informer's performance in a quantified manner.

3. It is necessary to compare with other state-of-the-art algorithms to demonstrate the performance of the proposed method.

4. Each variable in the formula needs to be explained.

5. page 5/8, before and ater Fig 3, redundant description.

'In this paper, three commonly used wavelet bases, Morlet wavelet base, Gaus

wavelet base, and Marr wavelet base, are selected to verify the timing feature capture

capability of CL-Informer, and the Gaussian wavelet base is selected as the eighth level

wavelet base in the Gaussian wavelet series, which contains a filter of length 8.'

'This study selected three commonly used wavelet bases, namely Morlet wavelet,

Gaus wavelet, and Marr wavelet, to verify their ability to capture temporal features in

CL-Informer. For Gaus wavelet, the eighth-level wavelet basis from the Gaussian

wavelet series was chosen, which includes a filter of length 8.'

6. Each image needs to be explained, clarifying what the symbols in the pictures represent.

7.The discussion section needs to include a quantitative analysis of the superiority of the proposed method and discuss which strategies have improved the performance of the algorithm.

8. The abstract and conclusion sections need improvement.

Reviewer #3: In the article, the authors propose a prediction model based on CL-Informer, which incorporates two improvements on the basis of the original Informer model: firstly, a wavelet transform embedding layer is added, where wavelet decomposition technique is first used to frequency split the original data, and then positional embedding is performed. This enhances the model's ability to analyze multi-scale data. Finally, the LSTM layer is used to replace the "distillation" in the original model to solve the redundant information generated by the wavelet transform operation. The article has some innovation, but the principle analysis of the model is not clear enough, and the research problem is not clear enough. Therefore, the following modifications are proposed:

1. The embedding layer based on wavelet transform is designed, it is obviou that the frequency division operation is carried out first, after the wavelet transform operation, how the data are input to the embedding layer of the model, please explain.

2. In the paper, we choose to remove the "distillation" operation of the Informer model and add the LSTM layer. The "distillation" operation in the Informer is to improve the training efficiency and generalization ability of the model, while adding the LSTM layer is to extract the dependency relationship between the data and the LSTM layer. The "distillation" operation in Informer is to improve the training efficiency and generalization ability of the model, while the LSTM layer is used to extract the dependency relationship between the data, the two roles are not the same, whether it is reasonable to replace the "distillation" operation of the Informer model with LSTM, please explain.

3. There is no specific formula for positional embedding in the embedding layer, please explain in detail the specific operation of the embedding layer.

4. Modify Figure 2, which does not clearly reflect the overall structure of the layer and the operation process.

5. The introduction of the CL-informer model proposed in the paper is not clear enough.

Reviewer #4: Overall, I found this paper to be quite good. It is not particularly well written and I would urge the authors to improve that aspect in many sections. Also, there is a fair amount of unexplained nomenclature and long sentences that could be summarized better with an equation. Overall, I think this paper demands minor modifications only.

more important points to clarify, modify:

- in general, most equations do not flow with the text. I.e. "The calculation process is illustrated by Eq.(1):", should be "The process can be written as, ..."

-Many datasets (ETTh2, ETTm1) are discussed in the wavelet selection part of the paper before being described. All datasets should be described before being used.

-Wavelet selection section. One of the main points of the paper is that wavelets are used. I would move this whole section as supplementary information and just say what wavelet is used for training and prediction but mention that interested readers can look up the SI where other wavelets are explored

-Wavelet selection - Give an equation for every wavelet so we can better understand the differences

more minor points

- intro - CNN undefined

- intro - remove of explain "However, the approximation method used in Nystromformer may

not be suitable for longer sequences"

- Problem definition - "The determination of whether to use single-element or

multi-element prediction depends on the value of dy ≥ 1". Remove, this was defined above the Eq.

- Eq.(5) is unnecessary since all contained in Eq.(4).

- do you mean integrating over the 'b' (if yes, please say): " By summing Wt

along the second dimension"

- Add an equation to describe this : " Finally, the multi-scale time-frequency domain

features are encoded by a convolutional block with convolutional kernels of size 3"

- There are an infinite number of wavelets, so this sentence does not make sense : "Due to the extensive experimental workload, not all wavelet bases"

were compared."

6. PLOS authors have the option to publish the peer review history of their article (what does this mean?). If published, this will include your full peer review and any attached files.

Reviewer #1: No

Reviewer #2: No

Reviewer #3: No

Reviewer #4: No

---

## [Author Response · Author response to Decision Letter 0]

28 Mar 2024

List of Responses

Dear Editor Aamna Mohammed AlShehhi, PhD and Reviewers：

Thank you for your letter and for the reviewers’ comments concerning our manuscript entitled “CL-Informer: Long Time Series Prediction Model Based on Continuous Wavelet Transform” (ID:PONE-D-24-03744). Those comments are all valuable and very helpful for revising and improving our paper, as well as the important guiding significance to our researches. We have studied comments carefully and have made correction which we hope meet with approval. Revised portion are marked in red in the paper. The main corrections in the paper and the responds to the reviewer’s comments are as flowing:

Responds to the reviewer’s comments:

Reviewer #1:

1.Response to comment:（Can the authors explain why they used these five time series datasets in this paper? So can CL-Informer handle bigger data?）

Response:Thank you very much for your professional review of our articles. The use of these 5 data sets is firstly to reflect the model's generalization in multiple fields. Secondly, ETTh1 and ETTh2 in ETT data adopt the same time granularity to verify whether the prediction effect of the model will be improved under the same type of data. ETTm1 verifies that time series prediction with different granularity in the same domain will also improve, while ECL and Wheather data verify the model's generalization in different domains. Larger data sets can be handled.

2.Response to comment: Why in S2_Table includes ETTh2, but in S1_Table doesn’t?

Response:Thank you very much for your question. ETTh2 is not included in S2_Table because ETTh1 and ETTh2 are in the same domain, with the same time granularity and the same type of data, so we did not make multivariate prediction for ETTh2 in multivariate prediction.

3.Response to comment: In Figure 3, the x-axis is the prediction length in dataset ETTh2, which is {24, 48, 168, 336, 720}, but in the main manuscript content, it says the ETTh2 data set with input length of 168 and prediction length of {24, 48, 96, 168, 720}, which one is correct?

Response:We are very sorry for our miswriting, but should have followed the predicted length of the picture as {24, 48, 168, 336, 720}. (Amended page 6, paragraph 2, line 7.)

4.Response to comment: From Fig 4, the authors claimed as the support length of the Gaus wavelet increases, the prediction performance gradually improves. For example, MSE from 0.06 (gaus1) to 0.05 (gaus8) is only 0.01 improved, it seems like not a strongly improved.

Response:We agree with the reviewer's opinion on this issue, but in fact, the MSE of CL-Informer gaus1 is 0.06445934, and the MSE of Gaus6 is 0.04488082, which has an increase of nearly 0.02. Moreover, each experiment was performed ten times, and the average MSE of the ten experiments of Gaus6 was about 0.02 higher than that of the ten experiments of Gaus1 wavelet. (We have redrawn Figure 4 to Figure 5 on page 7.)

5.Response to comment: Line 6 in paragraph one of the‘Results and analysis’ part needs to be modified, that sentence is not finished.

Response:We are very sorry for our careless mistake. Thank you for reminding me. We rewrote the unfinished part of the sixth line of "Results and analysis." (The changes are made in the second paragraph of "Results and analysis" on page 11, and lines 3 to 7 have been modified)

6.Response to comment:（Some figures can be merged as one figure, such as Figure 7 and Figure 8, the same datasets, but one is for MSE and the other is for MAE score, which can be put into one figure. The authors need to pay attention that in some figures they used MSE, but in others, they used MSE score.）

Response:We have corrected it according to the reviewer's comments. We merged Figure 7 and Figure 8 into Figure 8 and changed the 'MSE' and 'MSE score' in the legend to MSE. (Revised as Figure 8 on page 11)

7.Response to comment: It’s better to add some legends for the figures, like Figure 9, to make it more readable for the readers.

Response:We have corrected according to the reviewer's comments. We added a legend to Figure 9, adding a legend illustration of the X and y coordinates. (Revised as Figure 9, page 13).

8.Response to comment: It’s better to let a native speaker go through this whole manuscript and modify the grammar issues.

Response:Thanks for your suggestion. We have tried our best to polish the language in the revised manuscript.

Special thanks to you for your good comments.

Reviewer #2:

1.Response to comment: Quality of Fig 4,5,7, need improved 

Response:As suggested by the reviewer, we have redrawn Figure 4 and modified the quality of Figure 5 and Figure 7. (Modified as Figure 5 on page 7, Figure 6 on page 8, and Figure 8 on page 11)

2.Response to comment: In the abstract and conclusion, it is necessary to introduce the superiority of CL-Informer's performance in a quantified manner. 

Response:Considering the reviewers' suggestions, we quantitatively introduced the advantages of CL-Informer in the abstract and conclusion. (Changes are made to the 9th to last line of the summary on page 1 and the second paragraph of the conclusion on page 15.)

3.Response to comment: It is necessary to compare with other state-of-the-art algorithms to demonstrate the performance of the proposed method.

Response:We appreciate the reviewer's suggestion and compare MSE and MAE with other algorithms in Table 1 and Table 2.

4.Response to comment: Each variable in the formula needs to be explained.

Response:Thank you for your careful examination. We will add variables that are not explained in the formula.

5.Response to comment: page 5/8, before and ater Fig 3, redundant description.

Response:We apologize for our carelessness. In our resubmitted manuscript, the description of redundancy has been revised. Thank you for your correction.(The redundant description in Figure 4 (original Figure 3) has been deleted on page 6.))

6.Response to comment: Each image needs to be explained, clarifying what the symbols in the pictures represent.

Response:Thanks for your suggestion; we will add an explanation of the symbols in the picture.

7.Response to comment: The discussion section needs to include a quantitative analysis of the superiority of the proposed method and discuss which strategies have improved the performance of the algorithm.

Response:Thanks for your suggestion; we have added a quantitative analysis of the advantages of the proposed method in the discussion section (revised on page 14, lines 2 to 5).

8.Response to comment:（The abstract and conclusion sections need improvement.）

Response:We think this is an excellent suggestion. We introduced the advantages of CL-Informer by adding quantitative methods to the abstract and conclusion. (Changes are made to the 9th to last line of the summary on page 1 and the second paragraph of the conclusion on page 15.)

Special thanks to you for your good comments.

Reviewer #3:

1.Response to comment：The embedding layer based on wavelet transform is designed, it is obviou that the frequency division operation is carried out first, after the wavelet transform operation, how the data are input to the embedding layer of the model, please explain.

Response:We think this is a good suggestion. We added Figure 3 for the architecture diagram of the CWT embed layer and rewrote the CWT embed layer. (The changes are in Figure 3 on page 5 and the last paragraph on page 5)

2.Response to comment:Thank you for your advice. "Distillation operation" In this article, we removed the "distillation" operation from the Informer model and added the LSTM layer. Please indicate whether replacing the "distillation" operation of the Informer model with LSTM makes sense.

Response:Thank you for your advice. "Distillation operation" is deleted because although it improves the model's generalization ability, it will affect LSTM's extraction of features that are long dependent on time series during distillation operation. We once used a CL-Informer model that did not remove the "distillation" operation, and its MSE, MAE, showed a low MAE value for a CL-Informer model that did not remove the "distillation" operation. (Modified at lines 5-8, paragraph 2, page 2)

3.Response to comment: There is no specific formula for positional embedding in the embedding layer, please explain in detail the specific operation of the embedding layer.

Response:Thanks to your suggestion, we have rewritten the CWT embedding layer (changes are made in the last paragraph of page 5).

4.Response to comment: Modify Figure 2, which does not clearly reflect the overall structure of the layer and the operation process.

Response:Thanks for your suggestion; we have added the architecture diagram of the embedded layer in Figure 3-CWT to supplement the overall structure of the layer in Figure 2. (Revised as Figure 3 on page 5)

5.Response to comment: The introduction of the CL-informer model proposed in the paper is not clear enough.

Response:Thanks for your suggestions; we will try our best to improve our presentation of the model.

Special thanks to you for your good comments.

Reviewer #4:

1.Response to comment: in general, most equations do not flow with the text. I.e.

Response:Thanks to your suggestion, we have modified the fluidity of the equation.

2.Response to comment: Many datasets (ETTh2, ETTm1) are discussed in the wavelet selection part of the paper before being described. All datasets should be described before being used.

Response:We have made corrections based on the reviewer's comments. 2. Data ETTh1 and ETTm1 are described before wavelet selection. (Amended at page 6, paragraph 1, lines 3 to 7)2.Response to comment: Each image needs to be explained, clarifying what the symbols in the pictures represent.

3.Response to comment: Questions about wavelet selection and insert S1 as supplementary information.

Response:Thanks to the reviewer's suggestion, we slightly adjusted the part of the wavelet selection. We first added the architecture diagram of the CWT embedding layer in Figure 3 and rewrote the CWT embedding layer. Then, the choice of wavelet will be introduced to make the article smoother. (The changes are in Figure 3 on page 5 and the last paragraph on page 5)

4.Response to comment:"CNN" and "The approximate method used in Nyströmformer may not apply to longer sequences "are not defined.

Response:We have made corrections based on the reviewer's comments. We have a brief mention of CNN and a rewrite of the pros and cons of Nystromformer's long series predictions. (Changes are made on lines 1 to 2 and lines 22 to 24 of page 2)

5.Response to comment: Problem definition - "The determination of whether to use single-element or multi-element prediction depends on the value of dy ≥ 1". Remove, this was defined above the Eq.

Response:Thanks to the reviewer's suggestion, the multi-element prediction was removed depending on the value of dy≥1. (Modify the last paragraph of the Problem definition on the second page)

6.Response to comment: Formula (5) is not necessary because all are included in the formula (4), and an equation is added that describes "by summing over Wt along the second element."

Response:Thanks to the reviewer's suggestion, we deleted the original formula 5, added a new formula 5,6, and added the variable description. (Modified by Formula 5 on page 5 and Formula six on page 6)

7.Response to comment: The number of wavelets is infinite, so the sentence does not make sense: "Due to the extensive experimental workload, not all wavelets are based" on comparison.

Response:Thanks to the reviewer's suggestion, we deleted the section and rewrote it. (Amended first paragraph, page 15)

Special thanks to you for your good comments.

Other changes:

1.We made a structural change to the "Results and Analysis" section, bringing the original first paragraph under Figure 8. (Amended on page 11)

2.We changed the position of Table 1 under "Datum line" and put it into the "Cell time prediction:" of "Result and Analysis" to ensure a smoother article.(Revised at page 16)

3.We divide the first paragraph of "Results and Analysis" into two paragraphs. (Revised at page 15, paragraphs 1 and 2)

4.We are sorry for our carelessness. We have corrected the Informer model in Table 3 with MSE of 288 input length and 720 output length, and MAE of 576 input length and 1440 output length to be the same as the experimental data. (The Informer's input length is 288 with an output length of 720 MSE, and the MAE with an input length of 576 and an output length of 1440 has been corrected in Table 3 on page 14.)

We tried our best to improve the manuscript and made some changes. These changes will not influence the content and framework of the paper. Moreover, we did not list the changes here but marked them in red in the revised paper.

We appreciate the Editors/Reviewers’ warm work earnestly and hope the corrections will be approved.

Once again, thank you very much for your comments and suggestions.

---

## [Decision Letter · Decision Letter 1]

6 May 2024

CL-Informer: Long Time Series Prediction Model Based on Continuous Wavelet Transform

PONE-D-24-03744R1

Dear Dr. Baijin Liu,

We’re pleased to inform you that your manuscript has been judged scientifically suitable for publication and will be formally accepted for publication once it meets all outstanding technical requirements.

Kind regards,

Aamna AlShehhi, PhD

Academic Editor

PLOS ONE

Additional Editor Comments (optional):

Reviewers' comments:

Reviewer's Responses to Questions

**Comments to the Author**

1. If the authors have adequately addressed your comments raised in a previous round of review and you feel that this manuscript is now acceptable for publication, you may indicate that here to bypass the “Comments to the Author” section, enter your conflict of interest statement in the “Confidential to Editor” section, and submit your "Accept" recommendation.

Reviewer #1: All comments have been addressed

Reviewer #2: All comments have been addressed

Reviewer #4: All comments have been addressed

2. Is the manuscript technically sound, and do the data support the conclusions?

Reviewer #1: Yes

Reviewer #2: Yes

Reviewer #4: Yes

3. Has the statistical analysis been performed appropriately and rigorously? 

Reviewer #1: Yes

Reviewer #2: Yes

Reviewer #4: Yes

4. Have the authors made all data underlying the findings in their manuscript fully available?

Reviewer #1: Yes

Reviewer #2: Yes

Reviewer #4: Yes

5. Is the manuscript presented in an intelligible fashion and written in standard English?

Reviewer #1: Yes

Reviewer #2: Yes

Reviewer #4: Yes

6. Review Comments to the Author

Reviewer #1: I found out that all the authors have put considerable effort into addressing the reports of the referees. So I have no further questions, and this paper is ready to be accepted.

Reviewer #2: In the revised version of the submission, all comments have been addressed. The quality of the article has improved significantly.

Reviewer #4: The paper has been greatly improved. If anything, the other wavelets used for comparing against Morlet would benefit from being explained somewhere. Otherwise, the authors have made a significant effort to improve the paper and address my comments

7. PLOS authors have the option to publish the peer review history of their article (what does this mean?). If published, this will include your full peer review and any attached files.

Reviewer #1: **Yes: **Jing Jiang

Reviewer #2: No

Reviewer #4: No

---

## [Editor Report · Acceptance letter]

21 Jun 2024

PONE-D-24-03744R1 

PLOS ONE

Dear Dr. Liu, 

I'm pleased to inform you that your manuscript has been deemed suitable for publication in PLOS ONE. Congratulations! Your manuscript is now being handed over to our production team.

Kind regards, 

on behalf of

Dr Aamna AlShehhi 

Academic Editor

PLOS ONE